# μnit Scaling: Simple and Scalable FP8 LLM Training

**Saaketh Narayan** [1]  **Abhay Gupta** [2]  **Mansheej Paul** [1]  **Davis Blalock** [2]

## Abstract

Large language model training with 8-bit floating point (FP8) formats promises significant efficiency improvements, but reduced numerical precision makes training challenging. It is currently possible to train in FP8 only if one is willing to tune various hyperparameters, reduce model scale, or accept the overhead of computing dynamic scale factors. We demonstrate simple, scalable FP8 training that requires no dynamic scaling factors or special hyperparameters, even at large model sizes. Our method, *μnit Scaling (μS)*, also enables simple hyperparameter transfer across model widths, matched numerics across training and inference, and other desirable properties. μnit Scaling is straightforward to implement, consisting of a set of minimal interventions based on a first-principles analysis of transformer operations. We validate our method by training models with parameters ranging from 1B to 13B, performing all hidden linear layer computations in FP8. We achieve quality equal to higher-precision baselines while also training up to 33% faster.

## 1. Introduction

Because LLM training is computationally expensive, low-precision training provides large compute savings. Modern LLMs are typically trained in mixed-precision bfloat16 (BF16), where most computation occurs in BF16, but some components requiring higher precision (such as accumulators and master weights) use FP32 (Micikevicius et al., 2018). Thanks to increased hardware support for FP8 formats, mixed precision training using FP8 computation promises even greater training efficiency (Micikevicius et al., 2022). However, the reduced range and resolution of FP8 make LLM training challenging. In this work, we demon-

strate a simple, scalable FP8 training method with straightforward hyperparameter transfer on large LLMs, called "μnit Scaling" (μS).

Our μnit Scaling method builds on Unit Scaling (Blake et al., 2023), which aims to maintain unit variance in weights, activations, and gradients. To ensure this, it scales neural network operations with static constants and initializes network parameters to have unit variance. If all tensors used in training can maintain unit variance, they are representable with sufficient range and resolution by low-precision formats like FP16 and FP8. However, preserving high-quality tensor representations in low-precision formats is challenging for large models.

Besides faster training, several other properties are desirable in a low-precision training scheme. Examples include minimizing extra hyperparameters, avoiding dynamic scale factor overhead, and allowing optimal hyperparameters from small models to transfer to large models. As summarized in Fig. 1, μS is the only method that provides these benefits. We elaborate on each of these properties below.

**Straightforward hyperparameter transfer**   Tuning hyperparameters for large LLMs is expensive. A promising way to reduce this cost is to tune the hyperparameters for smaller LLMs and "transfer" them to large ones, either by using them directly or by applying a model-size-based formula as explored in μ-Parametrization (μP)(Yang et al., 2021; 2024a;b). However, applying hyperparameter transfer techniques in practice to low-precision training can be challenging; frequent divergences due to numerical issues may require training in higher precisions like FP32 (Yang et al., 2021). To address this, Blake et al. (2024) introduced u-μP, which combines Unit Scaling (Blake et al., 2023) and μP to enable hyperparameter transfer in low precision. Unfortunately, compared to conventional BF16 mixed precision training (henceforth termed "standard parametrized" (SP) models), both μP and u-μP have many more hyperparameters to sweep over (see Table 3), diminishing realized compute savings and increasing complexity. Specific implementation intricacies, such as zero-initialized queries in μP or LR scaling for embeddings by fan-out in u-μP, make these schemes harder to use in practice than SP. In contrast, our μnit Scaling (μS) scheme combines μP and Unit Scaling in a greatly simplified way, making it easier to use and more

[1]Work done while at Databricks Mosaic Research [2]Databricks Mosaic Research, San Francisco, CA. Correspondence to: Saaketh Narayan <narayan.saaketh@gmail.com>, Davis Blalock <davis.blalock@databricks.com>.

*Proceedings of the 42nd International Conference on Machine Learning*, Vancouver, Canada. PMLR 267, 2025. Copyright 2025 by the author(s).

| Method | Uses FP8 | Hparam transfer | Number of Hparams | No dynamic scaling factors | Scales stably to large models | Training-Inference precision match | Efficient distributed training |
|---|---|---|---|---|---|---|---|
| BF16 mixed precision (SP) | No | No | 3 | Yes | Yes | No | Yes |
| Maximal Update Parametrization (μP) | No | Yes | 6 | Yes | Yes | No | Yes |
| Unit Scaling / u-μP | Partially | Yes (u-μP) | 7 | Yes | Partially | Partially | Partially |
| Dynamically Scaled FP8 (SP), e.g. TE | Yes | No | 3 | No | Partially | Yes | Yes |
| **μnit Scaling (ours)** | **Yes** | **Yes** | **3** | **Yes** | **Yes** | **Yes** | **Yes** |

*Figure 1.* **Comparison of low-precision training methods.** Our proposed method, μnit Scaling (μS, bottom row), enables FP8 training and hyperparameter transfer at scale. Unlike existing methods, it does not use dynamic scaling, requires only a small set of hyperparameters, permits FP8 computation for all hidden layers, and makes the model more easily quantizable for inference.

cost-effective. We demonstrate hyperparameter transfer of learning rate ($\eta$) and weight decay ($\lambda$) to models of up to 20x larger widths.

**No Dynamic Scaling** With dynamic scaling, one calculates per-tensor scaling factors for each weight, activation, and gradient tensor in training. These scales shift BF16 tensors into the representable ranges of FP8 formats in each forward and backward pass. Typically, one also decouples the forward and backward formats, using e4m3 for weights and activations and e5m2 for gradients (Sun et al., 2019). NVIDIA's TransformerEngine is a notable example of an FP8 training library that uses dynamic scaling (NVIDIA, 2023). Calculating scaling factors dynamically adds training and inference overhead and complicates large-scale distributed training and checkpointing.

**Apply to All Linear Layers** Existing work on applying Unit Scaling at larger scales requires certain "critical matmuls" (attention out projection, FFN down projection) to stay in BF16 (Blake et al., 2024). Assuming a transformer model with conventional multiheaded attention and an MLP with an expansion ratio of 4, this means 41.7% of all hidden linear layer FLOPs are not in FP8. In contrast, μS ensures that, regardless of scale, *all* hidden layers use FP8.

**Match Inference-Time Quantization** For efficient inference, LLMs are often quantized to FP8 or INT8 for faster computation and reduced memory footprints (Khudia et al., 2021; Dettmers et al., 2022). Since training typically occurs in higher bitwidths (e.g., BF16), a mismatch in precisions at training time and inference time means that some level of quantization error is unavoidable, degrading model quality. Training with μS avoids this mismatch—since the LLM has already been trained in FP8, it is immediately ready for inference in FP8 for both weights and activations (W8A8).

### 1.1. Contributions

Our work makes the following contributions:

- Identifying root causes for poor numerics in conventional transformer blocks—for example, explaining diminishing variance in self-attention outputs with increasing sequence position.
- Introducing a simple method for fixing these issues that enables FP8 training in all hidden linear layers and with less overhead than existing methods. It also achieves desirable properties such as improved training efficiency and matched numerics at training and inference time.

## 2. Methods

In this section, we detail the components of our proposed method, μnit Scaling (μS). The modifications to the standard transformer training scheme that μS requires are summarized in Table 1. We elaborate on novel components such as our handling of self-attention numerics, residual modifications, and hyperparameter transfer below.

### 2.1. Self-attention Numerics

The causal self-attention mechanism at the core of decoder layers in LLMs is not variance-preserving, making low-precision training challenging.

Recall that standard self-attention is defined as:

$$\text{Attention}(\mathbf{Q}, \mathbf{K}, \mathbf{V}) = \text{softmax}\left(\frac{\mathbf{Q}\mathbf{K}^T}{\sqrt{d}}\right)\mathbf{V} \quad (1)$$

**Proposition 2.1.** *Suppose we have* $\mathbf{x} \in \mathbb{R}^k$ *and* $\mathbf{V} \in \mathbb{R}^{k \times m}$. *Define* $\mathbf{s} \triangleq \text{softmax}(\mathbf{x})$, $\mathbf{a} \triangleq \mathbf{s}^T\mathbf{V}$, *and* $\sigma_\mathbf{a}^2 \triangleq \text{Var}[\mathbf{a}]$. *Assume that each element* $x_i \overset{iid}{\sim} \mathcal{N}(0,1)$, *and that entries* $V_{ij}$ *are independent and distributed with* $\mu_\mathbf{V} \triangleq E[\mathbf{V}] = 0, \sigma_\mathbf{V}^2 \triangleq \text{Var}[\mathbf{V}] = 1$. *Then, up to a first-order Taylor approximation,* $\sigma_\mathbf{a}^2 \propto \frac{1}{k}$ *for* $k \gg 1$.

*Proof.* Recall that by the definition of the softmax function,

*Table 1.* **Components of the μS training scheme.** μS makes the following modifications to standard decoder-only transformer training practices. A deeper explanation of these modifications is provided in Appendix A.1.

| Modification | Description |
|---|---|
| Linear layer scaling factors | $\frac{1}{\sqrt{\text{fan\_in}}}$ static scaling factor applied in *both* forward and backward pass. The final LM head uses a multiplier of $\frac{1}{\text{fan\_in}}$ instead, in line with μP. |
| Res-Post-LayerNorm | LayerNorm is the last operation in each residual branch instead of the first. |
| "Fixed" residual modification | Use a fixed constant $\tau$ to make residuals variance-preserving, according to Eq. 11. |
| Unit variance initialization | All linear layer weights initialized with variance 1. |
| FP8 hidden layers | Use FP8E4M3 for weights and activations, FP8E5M2 for gradients. Before casting, clip BF16 values to FP8 dtype max. Keep embedding table and LM head in BF16. |
| Learning rate ($\eta$) scaling | Optimal $\eta$ stays constant for input and output layers, but is scaled by $\frac{\sqrt{d_{\text{base}}}}{\sqrt{d_{\text{model}}}}$ for all hidden layers, when transferring from a base model with width $d_{\text{base}}$ |
| Weight decay ($\lambda$) scaling | With fully decoupled weight decay, optimal $\lambda$ stays constant for all layers with increasing width. |

$s_i = \text{softmax}(\mathbf{x})_i = \frac{e^{x_i}}{\sum_{j=1}^{k} e^{x_j}}$. Denote the vector of elements' numerators $e^{x_i}$ as $\mathbf{n}$ and the vector of denominators $\sum_{j=1}^{k} e^{x_j}$ as $\mathbf{d}$, such that $\mathbf{s} = \frac{\mathbf{n}}{\mathbf{d}}$. Since $x_i \overset{\text{iid}}{\sim} \mathcal{N}(0,1)$, $\mathbf{n}$ is log-normally distributed and $\mathbf{d}$ is a sum of log-normals. This implies that[1]:

$$\mu_{\mathbf{n}} = e^{1/2}, \quad \sigma_{\mathbf{n}}^2 = e(e-1)$$
$$\mu_{\mathbf{d}} = ke^{1/2}, \quad \sigma_{\mathbf{d}}^2 = ke(e-1) \qquad (2)$$
$$\text{Cov}[\mathbf{n}, \mathbf{d}] = \sigma_{\mathbf{n}}^2 = e(e-1)$$

We can then use first-order Taylor approximations to estimate the moments of $\mathbf{s}$ as the ratio $\frac{\mathbf{n}}{\mathbf{d}}$, as shown in Casella & Berger (2002), to obtain:

$$\mu_{\mathbf{s}} = \text{E}\left[\frac{\mathbf{n}}{\mathbf{d}}\right] = \frac{\mu_{\mathbf{n}}}{\mu_{\mathbf{d}}} = \frac{1}{k} \qquad (3)$$

$$\sigma_{\mathbf{s}}^2 = \text{Var}\left[\frac{\mathbf{n}}{\mathbf{d}}\right] \approx \frac{\sigma_{\mathbf{n}}^2}{\mu_{\mathbf{d}}^2} + \frac{\mu_{\mathbf{n}}^2 \sigma_{\mathbf{d}}^2}{\mu_{\mathbf{d}}^4} - 2\frac{\mu_{\mathbf{n}}\text{Cov}[\mathbf{n},\mathbf{d}]}{\mu_{\mathbf{d}}^3}$$
$$= \frac{e-1}{k^2} - \frac{e-1}{k^3} \qquad (4)$$

Note that Eq. 3 holds exactly from the fact that all $k$ entries in $\mathbf{s}$ are positive and must sum to 1. Now, because each element $a_j = \sum_{i=1}^{k} s_i V_{ij}$, with independent entries $V_{ij}$, and with the fact that $\mu_{\mathbf{V}} = 0$ and $\sigma_{\mathbf{V}}^2 = 1$, the mean and variance of $\mathbf{a}$ can be determined as:

$$\mu_{\mathbf{a}} = \sum_{i=1}^{k} \mu_{\mathbf{s}}\mu_{\mathbf{V}} = 0 \qquad (5)$$

$$\sigma_{\mathbf{a}}^2 = \sum_{i=1}^{k} \sigma_{\mathbf{s}}^2\sigma_{\mathbf{V}}^2 + \sigma_{\mathbf{s}}^2\mu_{\mathbf{V}}^2 + \sigma_{\mathbf{V}}^2\mu_{\mathbf{s}}^2 = \frac{e}{k} - \frac{e-1}{k^2} \qquad (6)$$

The first term dominates for large $k$ and so $\sigma_{\mathbf{a}}^2 \sim \frac{1}{k}$. □

---

[1]See Appendix A.2 for the derivation of $\text{Cov}[\mathbf{n}, \mathbf{d}]$

In the causal self-attention operation shown in Eq. 1, the attention logits matrix $\frac{\mathbf{Q}\mathbf{K}^T}{\sqrt{d}}$ is causally masked such that the row of logits for a token at sequence position $k$ has length $k$. For a given token, by Prop. 2.1, the output of the self-attention operation will therefore have variance inversely related to that token's sequence position $k$. This causes tokens that appear later in the sequence to have much smaller variance than those that appear earlier, as shown in Fig. 2.

To address this issue, we make use of a basic property of the variance of linear combinations of independent random variables. With $\mathbf{a}(k)$ denoting the outputs of self-attention applied over a sequence of length $k$, the variance of $\mathbf{a}(k)$ (denoted $\sigma_{\mathbf{a}(k)}^2$) is the variance of a sum of $k$ random variables $\{X_i, \ldots, X_k\}$ with coefficients $\mathbf{c} \in \mathbb{R}^k$:

$$\text{Var}\left[\sum_{i=1}^{k} c_i X_i\right] = \sum_i c_i^2 \text{Var}[X_i] = \mathbf{c}^T\mathbf{v}, \qquad (7)$$

where $v_i \triangleq \text{Var}[X_i]$, and the equality holds if all $X_i$ are independent. If $\forall i\colon v_i = 1$, we further have $\sigma_{\mathbf{a}(k)}^2 = \|\mathbf{c}\|_2$.

Now recall that the softmax operation outputs positive coefficients $\mathbf{s}$ that sum to 1. This means that if we simply set coefficients $c_i = \sqrt{s_i}$, we obtain:

$$\sigma_{\mathbf{a}(k)}^2 = \|\mathbf{c}\|_2 = \sqrt{\sum_i c_i^2} = \sqrt{\sum_i s_i} = 1. \qquad (8)$$

That is, by taking the square root of attention scores, attention can be made variance-preserving for independent value tokens. This modification, which we term "Square-Root Softmax attention", is shown in Eq. 9. Square-Root Softmax attention is also easily implemented via modern

attention kernels like Flex-Attention (Dong et al., 2024).

$$\text{Attention}(\mathbf{Q}, \mathbf{K}, \mathbf{V}) = \sqrt{\text{softmax}\left(\frac{\mathbf{Q}\mathbf{K}^T}{\sqrt{d_k}}\right)}\mathbf{V} \quad (9)$$

In practice, standard self-attention does have diminishing $\sigma$ as sequence position increases; however, the observed variance is consistently higher than predicted by the above analysis of independent elements. This same effect is observed even when using Square-Root Softmax attention, causing observed $\sigma$ to increase over sequence position instead (Fig. 2).

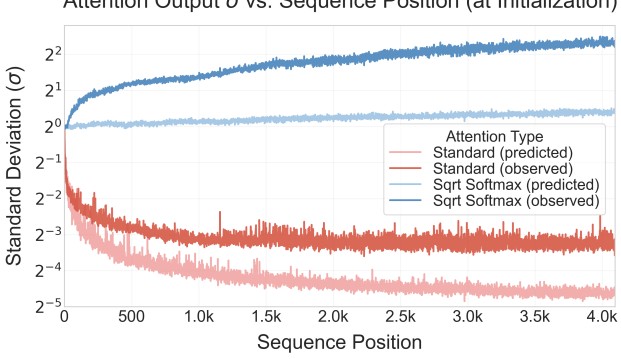

Figure 2. **Attention output variance changes over sequence length.** For standard attention, $\sigma$ decreases over sequence position both when simulated with iid value tokens (light red) and when observed in training (red). Taking the square root of attention scores keeps $\sigma$ constant when simulated with iid value tokens (light blue), but during training (blue), causes $\sigma$ to increase with sequence position. In practice, neither attention variant provides a consistent scale across outputs.

We provide a mechanistic explanation for this phenomenon: this increase in attention variance is an unavoidable consequence of the statistics of natural data. If all value tokens are truly independent, then Square-Root Softmax attention keeps $\sigma_{\mathbf{a}}$ constant. However, due to a high number of repeated tokens in real text data, value tokens are often highly correlated (Fig. 3). Due to this correlation, $\sigma_{\mathbf{a}}$ will be higher than predicted, and in the case of standard self-attention, diminish more slowly with respect to the token position.

To address this inconsistency in attention output variance, we use Res-Post-LayerNorm placement, as shown in Fig. 4(a). This architecture change consists of moving the normalization operation from the start of each residual branch to the end, and was first proposed in Liu et al. (2022) for training stability. Res-Post-LayerNorm ensures consistent $\sigma$ for all tokens in the residual stream, regardless of sequence position, correlation with other tokens, or the distribution of attention scores. A convergence test on 100-layer models validating the Res-Post-LayerNorm transformer against the standard Pre-LayerNorm transformer is

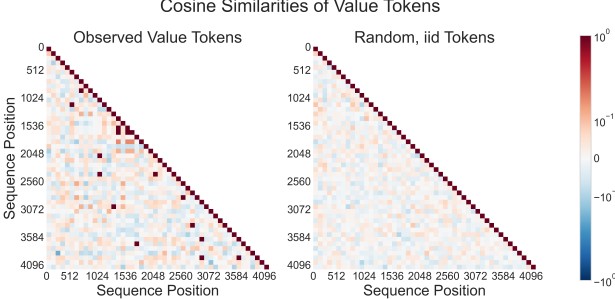

Figure 3. **Value tokens in text are highly correlated.** Comparison of cosine similarity between observed value tokens in a text data distribution versus value tokens $\overset{iid}{\sim} \mathcal{N}(0, 1)$. Repeated tokens in the value matrix, an unavoidable result of token frequency in real text data, lead to higher-than-random $\sigma$ as sequence position increases (cf. Fig. 2).

shown in Fig. 4(b). All μS models we train use Res-Post-LayerNorm.

## 2.2. Residual Modification Schemes

Every skip connection in a neural network adds another tensor to the residual stream. Summing all these tensors tends to increase the variance of the residual stream deeper in the network. To make residual connections variance-preserving instead, Blake et al. (2023) proposed replacing simple summation with weighted summation, where the weights $a$ and $b$ of the skip connection and residual branch satisfy $a^2 + b^2 = 1$. They proposed two methods for setting these coefficients: *fixed* and *running-mean*, which are shown in Eq. 11 and Eq. 12, respectively. The former uses a constant coefficient $\tau$, while the latter uses coefficients that are a function of the layer index $l$. The standard residual layer modification is shown in Eq. 10.

$$\text{standard} : x_{l+1} = x_l + f(x_l) \quad (10)$$

$$\text{fixed}(\tau) : x_{l+1} = \sqrt{1-\tau} \cdot x_l + \sqrt{\tau} \cdot f(x_l) \quad (11)$$

$$\text{running-mean} : x_{l+1} = \sqrt{\frac{l}{l+1}} \cdot x_l + \sqrt{\frac{1}{l+1}} \cdot f(x_l) \quad (12)$$

As shown in Fig. 5, we found that using either modification is better than the standard approach, with the *fixed* scheme providing better convergence than the *running-mean* scheme. All μS models we train therefore use the *fixed* scheme. We set the coefficient $\tau$ based on the depth using the results in Appendix A.3.

## 2.3. Hyperparameter Transfer with μnit Scaling

Zero-shot hyperparameter transfer allows hyperparameters to be tuned on a small proxy network, then directly used

(a)

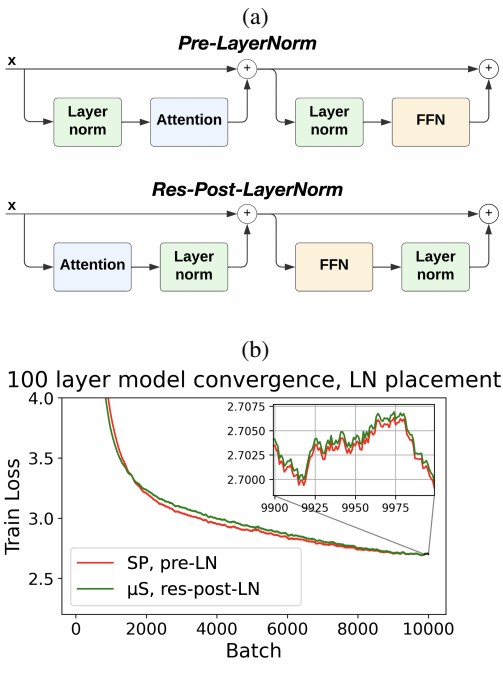

(b)

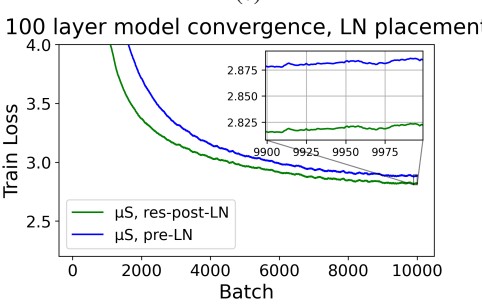

(c)

Figure 4. **Res-Post-LayerNorm.** **(a)** Pre-LayerNorm transformer architecture versus Res-Post-LayerNorm architecture. Res-Post-LayerNorm moves the LayerNorm operation from the start of each residual branch to the end (Liu et al., 2022). This ensures consistent variance across tokens when added to the residual stream. In contrast, Pre-LayerNorm networks permit unnormalized representations with inconsistent variance to be added to the residual stream, as shown with self-attention outputs in Fig. 2. **(b)** Convergence test loss curves with 100-layer models show that μS with Res-Post-LayerNorm achieves nearly identical convergence versus SP with Pre-LayerNorm. **(c)** Additional convergence tests with 100-layer models show that Res-Post-LayerNorm achieves better convergence over Pre-LayerNorm with μS.

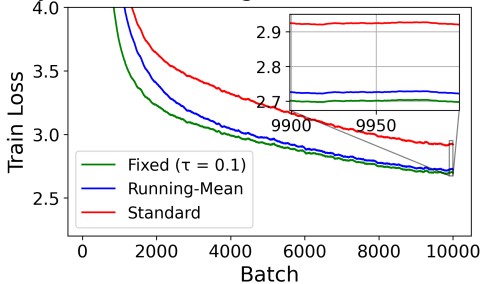

Figure 5. **Residual modification schemes affect μnit Scaled model convergence.** The *fixed* residual modification (green, Eq. 11) achieves better training convergence for deep transformers than the *running-mean* residual modification (blue, Eq. 12). The *fixed* residual coefficient for this model is $\tau = 0.1$. Both of these settings outperform the standard residual layer modification (red, Eq. 10).

on much larger networks without any further tuning (Yang et al., 2021). The width of the small proxy network is typically referred to as the "base width", or $d_{\text{base}}$. Because it eliminates the need to sweep hyperparameters at a large scale, such hyperparameter transfer yields massive compute savings.

Hyperparameter transfer with μnit Scaling follows from neural network equivalencies set forth in Yang et al. (2021, Appendix J.2.1), reproduced below for convenience. As detailed in Blake et al. (2024), Equations 13, 14, and 15 define the hidden layer in a model undergoing training. All hidden layers are initialized with weights $\mathbf{W}_0$ drawn from a normal distribution with variance $b^2$, use a learning rate of $c$, and have an output multiplier $a$. $\mathbf{X}$ and $\mathbf{Y}$ denote input and output activation matrices respectively; $t$ is the training time step; and $\mathbf{\Phi}_t(\nabla\mathcal{L}_0, \ldots, \nabla\mathcal{L}_t)$ denotes the weight update for time step $t$ using prior loss gradients.

$$\mathbf{W}_0 \sim \mathcal{N}(0, b^2) \tag{13}$$

$$\mathbf{Y} = a \cdot \mathbf{X}\mathbf{W}_t \tag{14}$$

$$\mathbf{W}_{t+1} = \mathbf{W}_t + c \cdot \mathbf{\Phi}_t(\nabla\mathcal{L}_0, \ldots, \nabla\mathcal{L}_t) \tag{15}$$

Under Adam-like optimizers, the output of this hidden layer is invariant to any scale factor $\theta > 0$ that changes $a, b, c$ as:

$$a \leftarrow a\theta, \quad b \leftarrow b/\theta, \quad c \leftarrow c/\theta \tag{16}$$

Under μP, $a = 1$, $b = \frac{1}{\sqrt{\text{fan\_in}}}$, and $c = \frac{1}{\text{fan\_in}}$. If we instead set $\theta = \frac{1}{\sqrt{\text{fan\_in}}}$, we obtain:

$$a = \frac{1}{\sqrt{\text{fan\_in}}}, \quad b = 1, \quad c = \frac{1}{\sqrt{\text{fan\_in}}} \tag{17}$$

Notice that $a = \frac{1}{\sqrt{\text{fan\_in}}}$ and $b = 1$ are exactly the output multiplier and unit initialization that Unit Scaling requires.

Therefore, the learning rate for hidden layers should scale as $\frac{1}{\sqrt{\text{fan\_in}}}$ for Unit Scaled models. This leads to the μS hyperparameter transfer scheme in Table 2.

In practice, given a base model with a width $d_{\text{base}}$, a new model with a width $d_{\text{new}}$, and optimal base model learning rate $\eta^*_{\text{base}}$, μS keeps $\eta^*_{\text{new}}$ constant for the embedding table, all LayerNorm parameters, and the LM head. The learning rate only changes for hidden layers, with $\eta^*_{\text{new}} = \eta^*_{\text{base}} \frac{\sqrt{d_{\text{base}}}}{\sqrt{d_{\text{new}}}}$.

*Table 2.* **μS scaling rules.** To transfer hyperparameters across model widths with μS, initialize layers, scale their outputs, and modify their learning rates as shown here.

|  | Weight Type | | |
|---|---|---|---|
|  | Input Layer | Final Layer | Hidden Layers |
| Init. Var. | 1 | 1 | 1 |
| Output Mult. | 1 | $1/\text{fan\_in}$ | $1/\sqrt{\text{fan\_in}}$ |
| Adam-like LR | 1 | 1 | $1/\sqrt{\text{fan\_in}}$ |

In addition to enabling hyperparameter transfer, μS also requires sweeping over a much smaller set of hyperparameters than existing schemes (Table 3).

*Table 3.* **Required hyperparameters in transfer schemes.** Hyperparameters used in practice to train transformer models under various schemes. While μP and related schemes provide better hyperparameter transfer than SP, they require sweeping over more hyperparameters to get reasonable model quality. In contrast, μS provides hyperparameter transfer and model quality with a much smaller set of hyperparameters. This makes the implementation simple and makes hyperparameter sweeps less expensive.

| Scheme | # Hparams | Hparams |
|---|---|---|
| **μS (ours)** | 3 | $\eta, \lambda, \tau$ |
| **SP** | 3 | $\eta, \lambda, \sigma_{\text{init}}$ |
| **μP** | 6 | $\eta, \lambda, \sigma_{\text{init}},$ $\alpha_{\text{res}}, \alpha_{\text{attn}}, \alpha_{\text{out}}$ |
| **u-μP** | 7 | $\eta, \lambda, \alpha_{\text{ffn-act}}, \alpha_{\text{attn-softmax}},$ $\alpha_{\text{res}}, \alpha_{\text{res-attn-ratio}}, \alpha_{\text{loss-softmax}}$ |

## 3. Results

### 3.1. Successful Hyperparameter Transfer

**Setup:** To evaluate hyperparameter transfer, we first train four-layer decoder-only LLMs with widths of 256 through 8192 using Standard Parametrization (SP) and μnit Scaling (μS). We begin with these small models since doing so allows us to collect ground truth optimal hyperparameters. All models use multi-headed attention (Vaswani et al., 2017) and were trained for 10,000 training steps with a

global batch size of 64 and sequence length of 1024 (i.e., 655M total tokens). SP models use Pre-LayerNorm placement and are trained in both BF16 and FP8 (using TransformerEngine). μS models were trained in both BF16 and FP8 and use Res-Post-LayerNorm placement (Fig. 4). μS used base models of width 256. For all models described in this and subsequent sections, we used the Lion optimizer (Chen et al., 2023) with fully decoupled weight decay and a cosine learning rate schedule decaying to 10% of the maximum learning rate. For details on why Lion is an Adam-like optimizer for hyperparameter transfer, please refer to Appendix A.4. All models were trained on Nvidia H100 GPUs using the Databricks MosaicML LLMFoundry (MosaicML, 2022a), Composer (MosaicML, 2021), and Streaming (MosaicML, 2022b) libraries.

**Hyperparameters:** We evaluate hyperparameter transfer over learning rate ($\eta$) and weight decay ($\lambda$). While μP Yang et al. (2021) does not give a theoretical basis for $\lambda$ transfer over width, we evaluate its transfer empirically because of its practical importance. Prior work by Lingle (2024) has shown that μP does not admit transfer of $\lambda$ with AdamW. However, Wang & Aitchison (2024) found that optimal $\lambda$ should scale with model size. To elucidate how $\lambda$ scales with model width, we jointly sweep over both $\eta$ and $\lambda$. We use fully decoupled weight decay, motivated by findings from Wortsman et al. (2024) that doing so results in more stable training. $\eta$ and $\lambda$ are swept over powers of 2. Based on the relationship between the residual coefficient $\tau$ and depth in Appendix A.3, the residual coefficient $\tau$ is 0.4 for these four-layer models.

As shown in Fig. 6, μS models have stable optimal learning rate ($\eta^*$) and weight decay ($\lambda^*$) from width 256 up to width 8192. Mirroring previous findings, $\eta^*$ for SP models decreases as the inverse of the width. $\lambda^*$ transfer across widths is relatively stable for both model types, with μS showing the most consistency.

### 3.2. FP8 Training at Scale

The previous section demonstrated hyperparameter transfer for small, shallow models. However, the real test of utility is scaling up to multi-billion-parameter models. This section demonstrates that μS allows us to train in FP8 while transferring hyperparameters for realistic model sizes. We also validate that our method is compatible with efficient distributed training.

**Setup:** We train 1B, 3B, 7B, and 13B parameter LLMs on approximately compute-optimal token budgets ($\sim$20x token-to-parameter ratio) using SP and μS, and in both BF16 and FP8, resulting in 4 individual models for each model size. The training configurations are detailed in Table 4. Based on the previous sections' hyperparameter transfer results (Fig. 6), we sweep $\eta$ and $\lambda$ on small models with a base

*Table 4.* **Large model training configurations.** Model training configurations for 1B, 3B, 7B, and 13B models. Only μS models use the residual coefficient $\tau$, which is dictated by model depth using results in Appendix A.3.

| Model | Params | Tokens | TPR | Steps | Batch Sz. | Seq. Len. | Width | Depth | # Heads | $\tau$ |
|---|---|---|---|---|---|---|---|---|---|---|
| **1B** | 1.6B | 31.5B | 19.4 | 7.5k | 1024 | 4096 | 2048 | 24 | 16 | 0.3 |
| **3B** | 3.0B | 62.9B | 20.8 | 15k | 1024 | 4096 | 2560 | 32 | 20 | 0.3 |
| **7B** | 7.3B | 140.0B | 19.3 | 16.7k | 2048 | 4096 | 4096 | 32 | 32 | 0.3 |
| **13B** | 13.6B | 260.1B | 19.1 | 31k | 2048 | 4096 | 5120 | 40 | 40 | 0.2 |

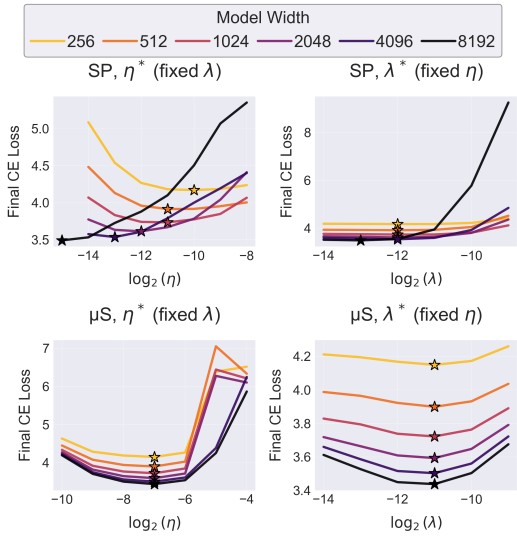

*Figure 6.* **With μS, optimal learning rate ($\eta^*$) and weight decay ($\lambda^*$) are stable across widths.** Optimal $\eta$ (left column) and $\lambda$ (right column) are shown across a range of model widths for models trained with SP (top row) and μS (bottom row). For each curve, the other hyperparameter is fixed at its optimal value. The base model width is 256. μS models have stable optimal $\eta$ and $\lambda$, even when width increases 32x to 8192. As expected, $\eta^*$ for SP models decreases with width. $\lambda^*$ is relatively stable as the width increases across both model types.

width of $d_{\text{base}} = 256$, then transfer optimal hyperparameters to large models with width $d_{\text{new}}$, as shown below.

- **SP:** all layers: $\eta^*_{\text{new}} = \eta^*_{\text{base}} \frac{d_{\text{base}}}{d_{\text{new}}}$, $\lambda^*_{\text{new}} = 0.5\lambda^*_{\text{base}}$

- **μS:** hidden layers: $\eta^*_{\text{new}} = \eta^*_{\text{base}} \frac{\sqrt{d_{\text{base}}}}{\sqrt{d_{\text{new}}}}$, $\lambda^*_{\text{new}} = \lambda^*_{\text{base}}$
  other layers: $\eta^*_{\text{new}} = \eta^*_{\text{base}}$, $\lambda^*_{\text{new}} = \lambda^*_{\text{base}}$

**Evaluation:** We use the Databricks Model Gauntlet to evaluate the quality of all models on specific tasks (Dohmann, 2023; Barton, 2024). These results are shown in Table 5.

We also compare model convergence via the final training cross-entropy loss averaged over the last 41.9M tokens (cor-

responding to 10 steps for 1B and 3B models and 5 steps for 7B and 13B models). Training loss curves are shown in Fig. 7.

As shown in Fig. 7, μS models train stably with FP8 even as the model size increases. We successfully transfer hyperparameters from a narrow base model with a width of 256 to models with widths up to 5120, demonstrating 20x width transfer (∼400x fewer FLOPs per run) in realistic, practical LLM training scenarios. This validates zero-shot hyperparameter transfer using μS. Evaluation results in Table 5 show that μS models achieve equal or better quality than SP models. These models demonstrate that μS successfully combines FP8 training with zero-shot hyperparameter transfer. To emphasize, all hidden layers use FP8 computation, and there are no dynamic scaling factors.

We also note that at the 13B scale, we attempted to remedy the divergence of the SP FP8 model by using multiple different values of $\lambda$, but this did not mitigate the frequent loss spikes and eventual divergence. μS models, by contrast, train stably. We also show the instability in training with Unit Scaling (US) at larger scales in Appendix A.5, motivating runs only with SP and μS for our final results.

### 3.3. FP8 Training Efficiency

To achieve state-of-the-art FP8 distributed training efficiency with μnit Scaling, we make use of operator fusion and static scaling. As shown in Fig. 8, FP8 training with μS is 25-33% faster than in BF16, and 1-6% faster than FP8 training with TransformerEngine (TE) (NVIDIA, 2023). All models were benchmarked on 64 NVIDIA H100 GPUs, and characteristics such as batch size and distributed training configuration were held constant. While TransformerEngine has fused modules such as LayerNorm-Linear or LayerNorm-MLP, we did not use those modules in order to make an equal comparison between μS and TE.

By relying on dynamic scaling, FP8 training with libraries like TE imposes additional overhead that is eliminated in μS. Calculating the absolute max of both the weight and activation tensors (or storing and reading past absolute max values in a delayed scaling approach) are operations that can be completely discarded in μS. Weights, activations, and gradients can be directly cast to FP8 formats, with a

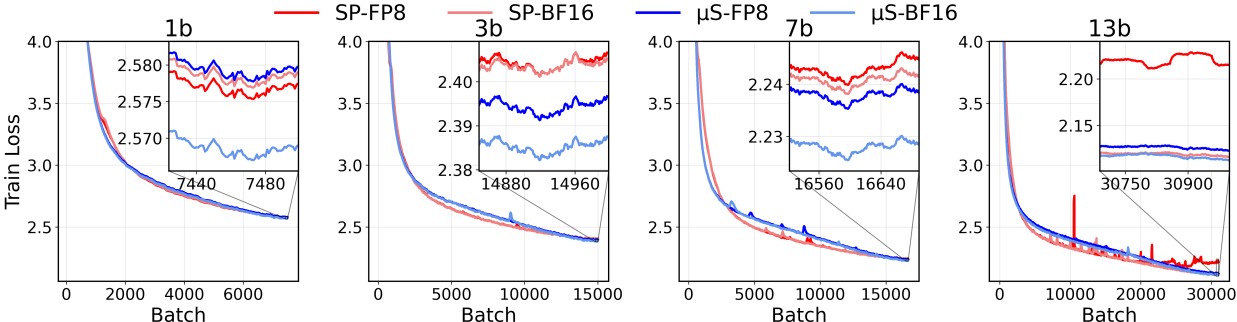

*Figure 7.* **µS models successfully train in FP8 at scale.** Comparison of training loss curves for standard parametrized (SP) and µnit scaled (µS) models in both FP8 and BF16, across 1B, 3B, 7B, and 13B parameter models. µS models successfully train in FP8 and converge to similar train loss values as their BF16 and SP counterparts. SP FP8 models are trained with TransformerEngine (TE). In our experiments at the 13B scale, SP models trained in FP8 with TE experienced frequent loss spikes and did not properly converge. We achieve state-of-the-art FP8 training efficiency via µS, with further details in Appendix 3.3.

*Table 5.* **Large model evaluation results.** We evaluate SP and µS models in FP8 and BF16 on a variety of tasks, with best results per eval and model size in bold. Final train loss (avg. over last ∼40M tokens) is also shown. µS models have equal or better quality than SP models, and maintain this quality even when training in FP8 as model size increases. Note that 13B SP FP8 models failed to properly converge, denoted by an asterisk.

| | 1B | | | | 3B | | | | 7B | | | | 13B | | | |
| | SP | | µS | | SP | | µS | | SP | | µS | | SP | | µS | |
| | BF16 | FP8 | BF16 | FP8 | BF16 | FP8 | BF16 | FP8 | BF16 | FP8 | BF16 | FP8 | BF16 | FP8* | BF16 | FP8 |
|---|---|---|---|---|---|---|---|---|---|---|---|---|---|---|---|---|
| Final Train Loss | 2.590 | 2.588 | **2.580** | 2.590 | 2.399 | 2.400 | **2.381** | 2.390 | 2.228 | 2.231 | **2.216** | 2.226 | 2.112 | 2.211 | **2.108** | 2.119 |
| ARC Easy (3-shot) | 52.1% | 52.4% | **53.4%** | 53.3% | 60.7% | 60.8% | **61.9%** | 60.8% | 67.2% | 65.6% | 67.1% | **68.0%** | **72.3%** | 35.7% | 71.8% | 69.7% |
| Jeopardy (3-shot) | 4.1% | 4.3% | **4.5%** | 3.5% | 13.4% | 11.3% | **16.8%** | 16.6% | 27.3% | 27.4% | **32.7%** | 30.6% | 40.2% | 0.2% | **43.1%** | 41.7% |
| SQuAD (3-shot) | 32.6% | **33.2%** | 30.9% | 31.3% | 42.3% | 45.3% | **47.9%** | 47.8% | 53.9% | 50.0% | **57.1%** | 55.1% | 52.9% | 1.5% | **62.8%** | 61.6% |
| HellaSwag (0-shot) | 47.2% | 47.5% | **48.3%** | 47.4% | 57.1% | 57.7% | **59.6%** | 59.5% | 66.8% | 66.5% | **69.2%** | 68.2% | 73.9% | 29.7% | **74.6%** | 74.3% |
| BIG-bench Wikidata QA (3-shot) | 47.3% | 48.6% | 49.3% | **50.2%** | 53.0% | 55.0% | 56.2% | **57.5%** | **60.4%** | 60.0% | 60.0% | 59.9% | **66.9%** | 4.0% | 66.1% | 62.9% |
| WinoGrande (5-shot) | **55.0%** | 52.6% | 51.1% | 52.0% | 58.8% | 54.9% | **59.5%** | 58.6% | 62.8% | 64.1% | **65.7%** | 65.3% | 70.3% | 57.8% | **71.1%** | 70.5% |
| OpenBookQA (10-shot) | **32.8%** | 32.4% | 32.0% | 32.4% | 37.8% | 38.2% | **38.8%** | 36.2% | 42.4% | 42.0% | **44.0%** | 41.8% | 45.2% | 26.6% | 45.8% | **46.6%** |
| PIQA (0-shot) | 70.7% | 71.1% | **71.5%** | 71.2% | 74.5% | **75.2%** | 74.3% | 74.3% | **77.2%** | 77.0% | 76.7% | 76.5% | 78.7% | 54.5% | **80.1%** | 79.4% |
| TriviaQA (3-shot) | 9.7% | 10.5% | **10.8%** | 9.7% | 17.8% | 17.7% | **20.4%** | 18.7% | 30.2% | 29.1% | 32.5% | **33.8%** | 42.4% | 0.5% | 44.3% | **44.8%** |
| Winograd (3-shot) | 64.5% | **69.6%** | 67.0% | 68.9% | 73.3% | 74.0% | 75.8% | **76.6%** | 78.8% | **80.6%** | **80.6%** | **80.6%** | 83.9% | 62.6% | **86.1%** | 82.8% |
| LAMBADA (0-shot) | **44.8%** | 44.5% | 43.6% | 41.3% | 52.8% | 54.2% | 55.9% | **57.4%** | 60.3% | 60.7% | 63.0% | **64.6%** | **65.7%** | 34.8% | 61.6% | 64.3% |
| CoQA (0-shot) | 19.3% | **21.3%** | 20.8% | 20.0% | 26.2% | 25.4% | 27.9% | **28.6%** | 28.2% | 32.0% | 33.3% | **35.0%** | 39.8% | 13.2% | 44.4% | **44.6%** |
| ARC Challenge (3-shot) | 25.4% | 26.0% | **27.8%** | 25.0% | 30.3% | 30.1% | **31.8%** | 30.9% | 36.1% | 35.7% | 38.3% | **39.0%** | 42.0% | 27.6% | **42.2%** | 41.5% |
| COPA (0-shot) | 65.0% | 68.0% | 64.0% | **70.0%** | 69.0% | 68.0% | 68.0% | **71.0%** | 76.0% | 76.0% | 78.0% | **80.0%** | 83.0% | 62.0% | **84.0%** | 78.0% |
| BIG-bench Operators (3-shot) | 12.4% | 12.9% | 13.8% | **14.3%** | **19.5%** | 17.1% | 17.1% | 18.6% | 21.4% | 20.0% | 20.0% | **23.3%** | 31.4% | 24.3% | **37.6%** | 37.1% |
| GSM8K (0-shot) | 2.4% | **2.6%** | 2.4% | 2.4% | **3.7%** | 1.7% | 2.3% | 2.0% | 3.9% | **5.0%** | 4.0% | 3.9% | 8.7% | 0.0% | 9.3% | **10.9%** |

constant $\alpha = \frac{1}{\sqrt{\text{fan\_in}}}$ scaling factor used in the hidden linear layers' GEMM calls, where a GEMM is defined as:

$$\mathbf{C} \leftarrow \alpha\mathbf{AB} + \beta\mathbf{C} \quad (18)$$

NVIDIA's H100 GPUs support FP8 GEMMs through the $cublasLtMatmul()$ operation (NVIDIA Corporation, 2024).

To maximize training speed and mirror TransformerEngine NVIDIA (2023), we fuse clipping to the FP8 range, casting to FP8, and transposing into a single Triton (Tillet et al., 2019) kernel. A transpose is necessary because H100s only support one layout ("TN") with FP8, but the forward and backward passes use different layouts (thanks to using $\mathbf{W}$ vs $\mathbf{W}^T$).

## 4. Conclusion

This work presents µnit Scaling (µS), an LLM training method enabling both statically-scaled FP8 computation and zero-shot hyperparameter transfer at scale. µnit Scaling consists of a set of principled model and optimization modifications, including Res-Post-LayerNorm, variance-preserving skip connections, unit-variance initialization, and straightforward scaling of optimization hyperparameters with model width. Compared to alternatives, µnit Scaling is simpler, faster, more stable across model scales, and has fewer hyperparameters. We demonstrate successful FP8 training with hyperparameter transfer at scale with high-quality µnit Scaled LLMs at 1B, 3B, 7B, and 13B sizes.

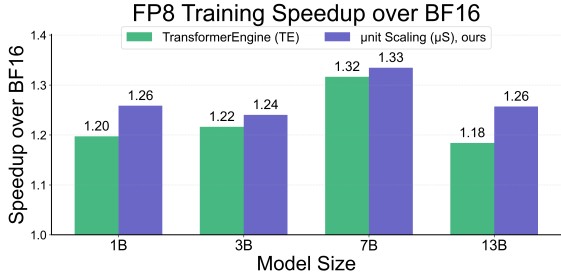

**Figure 8. Training in FP8 with μS achieves state-of-the-art efficiency.** FP8 training with μnit Scaling provides 25-33% higher throughput than BF16 training and 1-6% higher throughput than FP8 training with TransformerEngine (TE), over 1B, 3B, 7B, and 13B model sizes. Models are configured as specified in Table 4 and benchmarked on 64 NVIDIA H100 GPUs. Static scaling, operator fusion, and simplifications to Unit Scaling make this efficiency possible.

## Impact Statement

This paper introduces μnit Scaling (μS), a method designed to enhance the efficiency of Large Language Model (LLM) training through scalable FP8 computation and straightforward hyperparameter transfer. The advancements provided by μS could reduce both the computational and environmental costs associated with training large-scale models, potentially democratizing access to high-performance machine learning by lowering resource requirements. While this work's primary goal is advancing training efficiency, we acknowledge that, as with all machine learning technologies, continued attention to ethical considerations and societal implications remains important.

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

# A. Appendix

## A.1. Why these modifications?

Table 1 contains a number of modifications to standard bf16 training setups. Where did these come from? Are they simply a result of trying ideas until something worked? Or are they the result of more principled analysis and ablations?

While we do explain the basis for each modification over the course of the main text, this section summarizes how we arrived at each of them. We can group the origins of these changes into three categories: **simple math**, **adhering to prior art**, and **ablation experiments**.

### A.1.1. SIMPLE MATH

Recall that, in order to ensure stable training and consistent hyperparameter meanings, we wish to ensure that all weight and activation tensors have unit variance. Enforcing unit variance is difficult because the weights are constantly being modified throughout training. To enforce *exact* unit variance everywhere would require significant overhead in the form of added normalization operations. We therefore relax the constraint to the following:

1. Each residual branch must have exactly unit variance

2. Weight tensors must have unit variance at initialization

3. Linear layer outputs have unit variance at initialization, assuming the inputs are iid with unit variance.

4. Weight updates should attempt to preserve the weight and activation variances to the extent that this is possible without significant overhead.

The last three requirements mirror Blake et al. (2023) while the first is stronger.

Our core modifications follow immediately from these requirements and a bit of math.

**Unit variance initialization, linear layer scaling factors.** Suppose we initialize our weights with unit variance to achieve requirement (2). Given iid standard normal input elements, our outputs will be $\chi^2$ random variables with $k$ degrees of freedom, where $k$ is the contraction dimension. This has a mean and variance of fan_in and variance of $2 * \text{fan\_in}$, which are nowhere near 1 and so violate requirement (3). The typical solution to this is scaling down the initialization by a factor of $\sqrt{\text{fan\_in}}$, but this violates requirement (2). As observed in (Blake et al., 2023), we can reconcile both by scaling down the outputs by $\sqrt{\text{fan\_in}}$ *at runtime* as part of the GEMM call. This one extra multiply per output element is essentially free, and in fact fused into instructions such as the NVIDIA Hopper architecture's wgmma (NVIDIA). See Blake et al. (2023) for further discussion.

**Learning rate scaling.** Recall from (Yang et al., 2021) and Section 2.3 that one can scale weight initialization variance, learning rate, and linear layer output arbitrarily as long as all three are scaled according to a precise relationship. Since we have fixed the weight initialization variance to 1 and the output scaling to $\text{fan\_in}^{-\frac{1}{2}}$, our learning rate scale of $\text{fan\_in}^{-\frac{1}{2}}$ is uniquely determined. Further, when changing fan_in from $d_{base}$ to $d_{new}$, this implies scaling the learning rate by $\frac{\sqrt{d_{\text{base}}}}{\sqrt{d_{\text{new}}}}$.

### A.1.2. ADHERING TO BEST PRACTICES.

Some aspects of our training recipe are crucial but already common (though not universal) practices. These include:

**Weight decay ($\lambda$) scaling.** Recall that decoupled weight decay amounts to multiplying weights by a constant $1 - \lambda, 0 <= \lambda < 1$ during each update. This operation already has the same semantics across model widths.

**FP8 hidden layers.** Using e4m3 weights and activations along with e5m2 gradients is a common practice (NVIDIA, 2023; Micikevicius et al., 2022) Clipping instead of overflowing prevents NaN/Inf values. Keeping the first and last layers in higher precision is also common.

### A.1.3. ABLATION EXPERIMENTS.

Two modifications in our recipe can be implemented in multiple ways, so we chose the details based on smaller-scale experimental results.

*Table 6.* **Comparing μS with other schemes** μS components have commonalities and differences with existing training schemes. It is the only one which combines scalable, complete FP8 LLM training with hyperparameter transfer; see Figure 1 for a comparison of features of low-precision training methods.

| μS Component | μP | Unit Scaling | u-μP |
|---|---|---|---|
| Linear layer scaling factors | Not used | Used, but can be different in forward and backward pass. | Used |
| Res-Post-LayerNorm | Not used | Not used | Not used |
| "Fixed" residual modification | Not used | Proposed | Not used |
| Unit variance initialization | Not used | Used | Used |
| FP8 hidden layers | Not used | Used, but not at scale | Used, but restricted only to some layers |
| Learning rate ($\eta$) scaling | Used | Not used | Used |
| Weight decay ($\lambda$) scaling | Not used | Not used | Used |

**Fixed residual modification.** In order to satisfy our design goal of having a fixed-variance residual stream, we need to combine the previous residual stream tensor and the latest residual branch output in some manner that preserves variance. As discussed in the paper, this can be done by replacing summation with weighted summation. However, we are left with a degree of freedom in setting the weighting coefficient. To keep the search space small, we consider only the two schemes from (Blake et al., 2023) and decide between them based on the experiments in Section A.3.

**Res-Post-LayerNorm.** As we show in Section 2.1, the variance of token representations tends to collapse later in the sequence. If a closed-form correction could exactly undo this effect, we could apply such a correction and avoid modifying the architecture. However, as shown in Figures 2 and 3, the pattern of variance collapse is input-dependent and deviates greatly from what iid assumptions would lead one to expect. In order to satisfy our requirement that residual streams have unit variance, we therefore must resort to a blunt instrument: imposing normalization at runtime. We could normalize the residual stream itself, add a normalization op at the end of each residual branch, or move the normalization in a Pre-LN transformer from the start of the branch to the end. We decided to go with the last option because it adds no extra operations, normalizes both the residual stream token embeddings and their updates, is consistent with previous work (Liu et al., 2022; OLMo et al., 2024), and worked well in our ablation experiments (Fig 4b).

A.1.4. COMPARISON TO EXISTING SCHEMES

As a supplement to to Table 1 which enumerates the components of μS compared to standard practice (SP), Table 6 compares these components with μP, Unit Scaling, and u-μP.

**A.2. Covariance of softmax numerator and denominator**

In the proof for Prop. 2.1, we state that $\text{Cov}[\mathbf{n}, \mathbf{d}] = \sigma_{\mathbf{n}}^2$. Here we derive this result. Just as in Sec. 2.1, define $\mathbf{s}$ as the output of the softmax function applied to a vector of $k$ independent elements $\mathbf{x}$. The softmax function is defined as $s_i = \text{softmax}(\mathbf{x})_i = \frac{e^{x_i}}{\sum_{j=1}^{k} e^{x_j}}$. As shown previously, we denote the vector of elements containing numerators of elements of $\mathbf{s}$ as $\mathbf{n}$ and denominators of elements of $\mathbf{s}$ as $\mathbf{d}$, such that $\mathbf{s} = \frac{\mathbf{n}}{\mathbf{d}}$. By the definition of covariance:

$$\text{Cov}[\mathbf{n}, \mathbf{d}] = \text{E}[(n_i - \mu_{\mathbf{n}})(d_i - \mu_{\mathbf{d}})] \tag{19}$$

By the definition of softmax, $d_i = \sum_{i=1}^{k} n_i$, and by linearity of expectation, $\mu_{\mathbf{d}} = k\mu_{\mathbf{n}}$. Using this, we obtain:

$$\text{Cov}[\mathbf{n}, \mathbf{d}] = \text{E}[(n_i - \mu_{\mathbf{n}})(n_1 + n_2 + \ldots + n_i + \ldots + n_k - k\mu_{\mathbf{n}})] \tag{20}$$

Expanding this expression:

$$\text{Cov}[\mathbf{n}, \mathbf{d}] = \text{E}[(n_i - \mu_{\mathbf{n}})((n_1 - \mu_{\mathbf{n}}) + (n_2 - \mu_{\mathbf{n}}) + \ldots + (n_i - \mu_{\mathbf{n}}) + \ldots + (n_k - \mu_{\mathbf{n}}))] \tag{21}$$

By linearity of expectation:

$$\text{Cov}[\mathbf{n}, \mathbf{d}] = \text{E}[(n_i - \mu_{\mathbf{n}})^2] + \sum_{j \neq i} \text{E}[(n_i - \mu_{\mathbf{n}})(n_j - \mu_{\mathbf{n}})] \tag{22}$$

Because elements of the softmax input $\mathbf{x}$ are independent, and $n_i = e^{x_i}$, elements of $\mathbf{n}$ are also independent. Therefore $E[(n_i - \mu_{\mathbf{n}})(n_j - \mu_{\mathbf{n}})] = 0$ for $j \neq i$. Then by the definition of variance as $\text{Var}[\mathbf{n}] = E[(n_i - \mu_{\mathbf{n}})^2]$, we obtain:

$$\text{Cov}[\mathbf{n}, \mathbf{d}] = \text{Var}[\mathbf{n}] \tag{23}$$

### A.3. Modifying Residual Connections with $\tau$

To make skip connections variance-preserving, we use the *fixed* residual modification scheme, as shown in Eq. 11, with coefficients based on the hyperparameter $\tau$ (Blake et al., 2023). To understand the relationship of the optimal residual coefficient $\tau^*$ with network depth, we swept over various values of $\tau$ for models of different widths (256, 512, 1024, 2048) and depths (20, 40, 60, 80, 100). In order to assess potential confounding effects between $\tau^*$ and $\eta^*$ and $\lambda^*$, we tuned those two hyperparameters as well. We trained each model for 10.5B tokens with a global batch size of 256 and sequence length of 4096. We define the optimal subset of models as those which had a final cross-entropy loss within 0.25% of the optimum (with loss averaged over the last 10 steps, i.e. 10.5M tokens). As shown in Fig. 9, $\tau^*$ (for the optimal subset of models) decreases as network depth increases. Since the contribution of each residual branch exponentially decays with depth, a lower $\tau$ ensures a lower rate of decay, likely useful as networks get deeper. This relationship between $\tau^*$ and depth is consistent even as model width increases. In our experiments, $\tau$ can be coarsely swept. We use the results shown in Fig. 9, to directly choose $\tau^*$ for all μS model training.

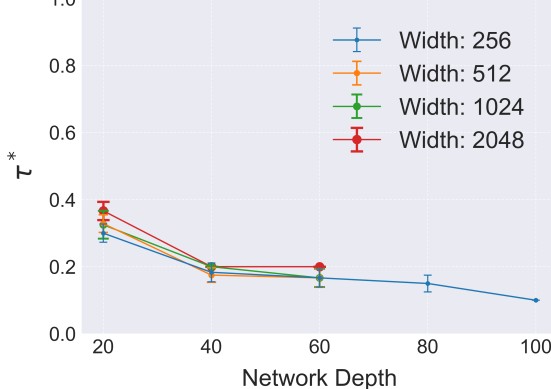

*Figure 9.* **Optimal residual coefficient $\tau^*$ decreases with depth.** The 3 hyperparameters of $\tau$, $\eta$, and $\lambda$ are swept for models of varying widths (256, 512, 1024, 2048) and depths (20, 40, 60, 80, 100). The mean and standard error of $\tau$ is shown for the optimal subset of models from each hyperparameter sweep, where a model is included in the optimal subset if it had final cross-entropy loss within 0.25% of the sweep optimum. $\tau^*$, which controls the decay rate of residual branch contributions in the residual stream, decreases as network depth increases.

### A.4. Lion Optimizer and Hyperparameter Transfer

Here, we show why Lion Chen et al. (2023) is an "Adam-like" optimizer, so the μP rules for hyperparameter transfer with Adam (Kingma & Ba, 2017) are applicable to Lion as well. Because Adam and Lion are both adaptive optimizers that normalize gradients coordinatewise before updating parameters, the nonlinear tensor product matrix results obtained in Yang et al. (2021, Appendix J.1.3) apply to both optimizers. One can see that Lion differs from Adam only in that it has a different second moment estimate. Under both optimizers, with gradient $g_t$, a parameter $\theta$ is updated as:

$$\theta_{t+1} = \theta_t - \eta \frac{\beta_1 m_t + (1 - \beta_1)g_t}{\sqrt{s_t}} \tag{24}$$

For Lion, this follows by expressing $\text{sign}(c_t)$ as $c_t/c_t^2$. Then, the second moment estimate $s_t$ for Adam (Eq. 25) and Lion (Eq. 26) are below.

$$s_t^{\text{Adam}} = \beta_2 v_t + (1 - \beta_2)g_t^2 + \epsilon \tag{25}$$

$$s_t^{\text{Lion}} = c_t^2 = \beta_1^2 m_t^2 + 2\beta_1(1 - \beta_1)m_t g_t + (1 - \beta_1)^2 g_t^2 \tag{26}$$

This justifies why Lion is an Adam-like optimizer for the purposes of hyperparameter transfer. We use Lion for its reduced memory footprint in all our experiments.

### A.5. μnit Scaling vs Unit Scaling for larger model training

We test the unit scaling (US) and μnit scaling (μS) methods at the 7B model scale with FP8 training. Figure 10 shows that unit scaling models diverge very early in training, while μnit scaling runs converge smoothly. Based on this experiment, we did not conduct final model runs at different model scales with unit scaling (1B–13B).

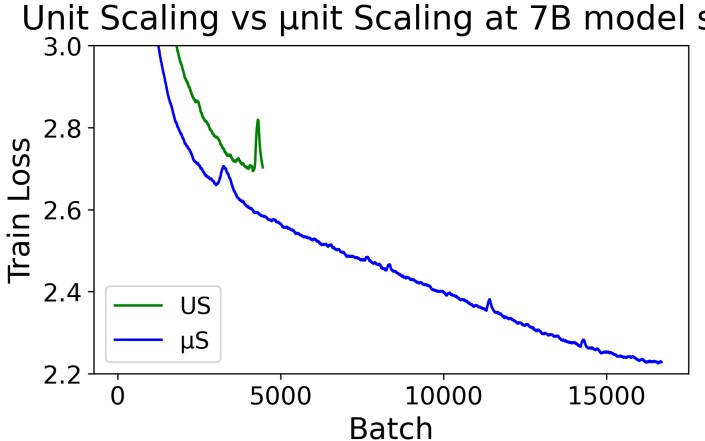

*Figure 10.* **Unit Scaling (US) vs μnit Scaling (μS) for 7B models.** Convergence test loss curves at 7B model scale show that μS converges smoothly while US training diverges early in training.

### A.6. Activation Outliers

We analyze activation distributions taken over 32,768 tokens at every 10 layers for all FP8 models trained according to Table 4, with results shown in Fig. 13. These figures show the distribution of activation values for attention and FFN block inputs and outputs in the final 1B, 3B, 7B, and 13B FP8 models. While SP models consistently have outliers in the attention block and FFN block inputs at all model scales, μS models do not have these outliers in block inputs. This may make μS models more easily quantizable. It is important to note, however, that in SP models, the Pre-LayerNorm placement means that activations from the residual stream are first normalized before subsequent operations.

While we do not identify the exact mechanism by which these outliers arise in the residual stream in SP models, we show their absence in μS models here, with activation distributions that may be more conducive to quantization. An activation distribution with fewer outliers requires fewer bits to represent it.

### A.7. Activation Function Choice

The choice of activation function can have a significant impact on activation underflow when training in FP8. For example, recent work by (Anonymous, 2024) identifies outlier amplification from SwiGLU as a challenge for FP8 LLM training. Nearly all state-of-the-art LLMs today use either SiLU or GELU as their activation function, but when training in FP8, this may lead to underflow in activations during training. This is because these functions asymptotically approach zero as inputs $x \to -\infty$. We define the FP8 underflow fraction, or the fraction of elements flushed to 0 from a BF16 to FP8 cast, as a metric to evaluate various activation functions. As shown in Fig. 11, this can cause many activations to underflow.

To better understand how activation function choice influences FP8 underflow when training with μnit scaling, we train small 4 layer models with GELU, SiLU, and ReLU. Our findings, detailed in Fig. 12 that during unit scaled model training, the choice of activation function drastically impacts the FP8 underflow rate for activation outputs. GELU greatly degrades

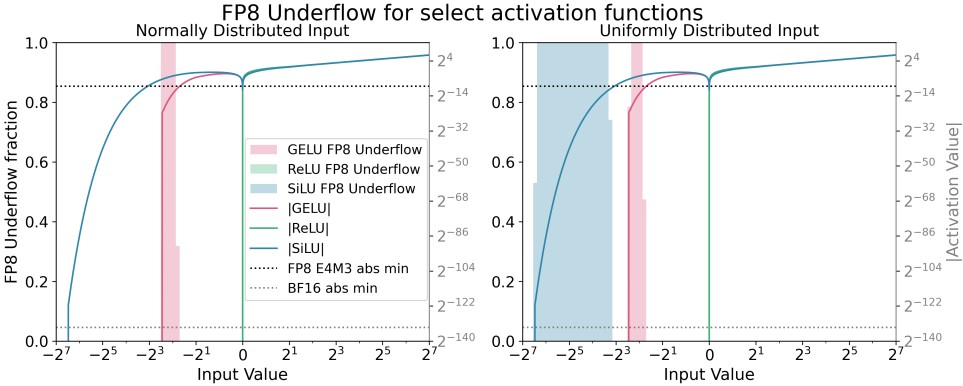

Figure 11. **Different activation functions cause different amounts of FP8 underflow.** When casting $\mathcal{N}(0, 1)$ or $\text{Unif}(-128, 128)$ values from BF16 to FP8 (e4m3), GELU, SiLU, and ReLU (green) erroneously round to zero (underflow) with different probabilties. GELU and SiLU experience significant FP8 underflow because they slowly approach 0 for increasingly negative inputs. SiLU approaches 0 more slowly than GELU and so underflows for a wider range of inputs. ReLU simply maps all negative values to 0, regardless of the numerical format.

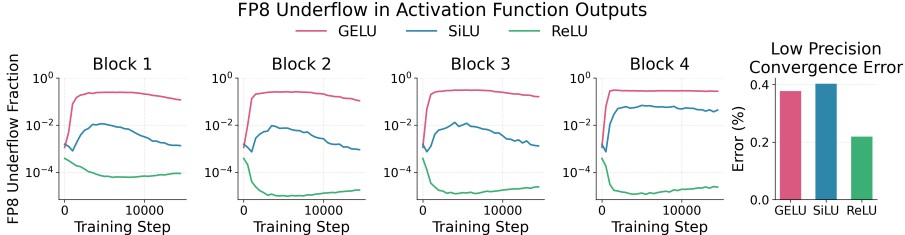

Figure 12. **Activation function choice impacts FP8 underflow and low-precision convergence error.** FP8 underflow of activation function outputs for each block in a 4 layer transformer model during training is shown for GELU, SiLU, and ReLU. Low precision convergence error, defined as the percent difference in final cross entropy loss between an FP8 model and its BF16 counterpart, is shown in the rightmost chart. GELU and SiLU cause significant underflow over the course of training, and models trained with these activation functions have twice as much low precision convergence error as with ReLU. ReLU greatly reduces this FP8 underflow by multiple orders of magnitude.

the representation of FFN down projection inputs, reaching up to 30% underflow during training. SiLU causes similar degradation, but at a lower rate, reaching up to 7% during training. In contrast, ReLU does not suffer from this problem, with a maximum of 0.04% FP8 underflow during training. As a result, FP8 unit scaled models trained with ReLU have smaller low-precision convergence error (defined as the percent difference between the final cross entropy loss an FP8 model and its BF16 counterpart). Based on these observations and results, ReLU minimizes FP8 underflow and low-precision convergence error. ReLU also has the added benefit of sparsifying activations, a property which enables significant inference-time optimizations (Mirzadeh et al., 2024). However, using GELU results in models with lower final training loss. For this reason, we use GELU when training all µS models. Additional investigations into activation functions more suitable for FP8 training can help mitigate underflow while also providing improved convergence.

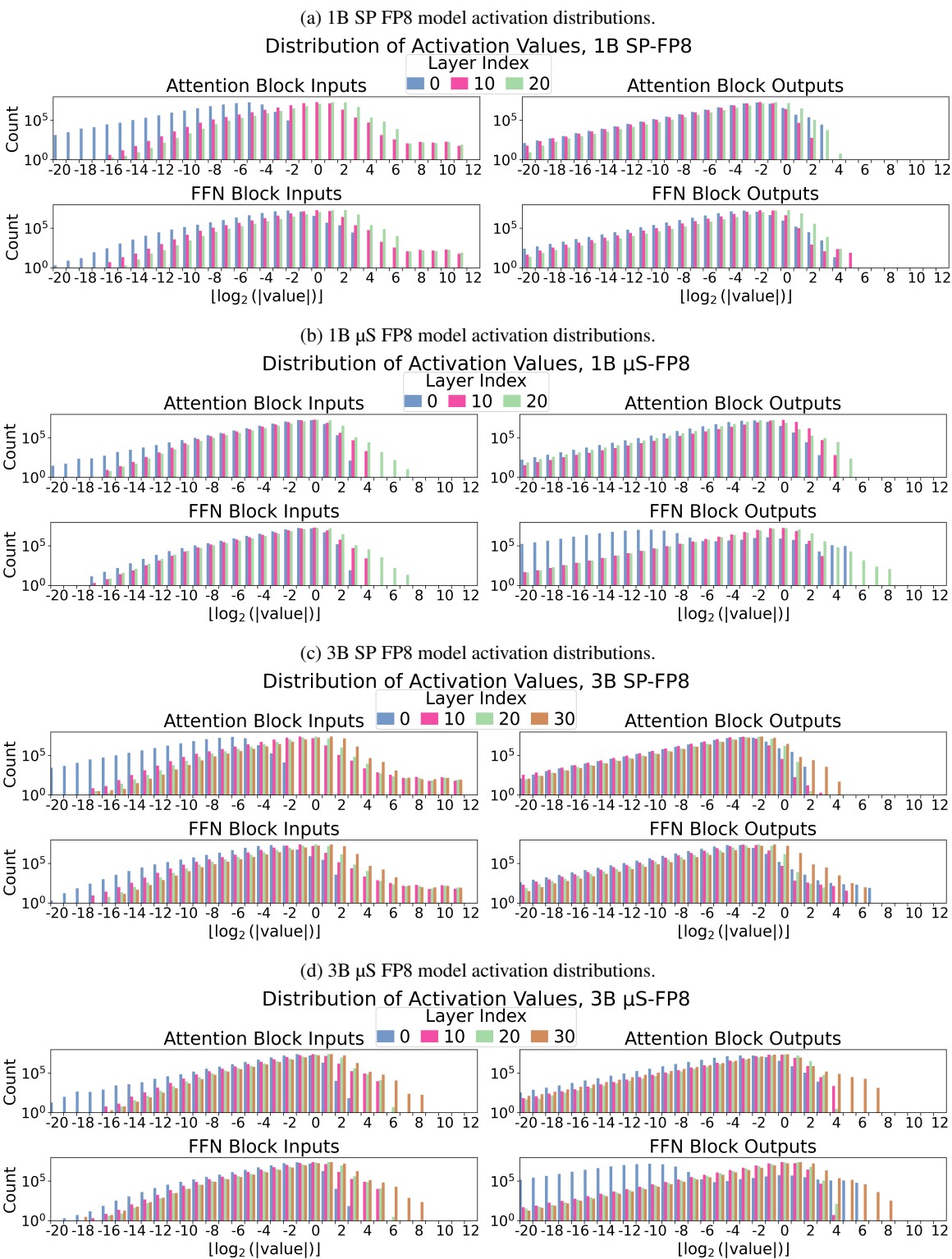

(a) 1B SP FP8 model activation distributions.

(b) 1B μS FP8 model activation distributions.

(c) 3B SP FP8 model activation distributions.

(d) 3B μS FP8 model activation distributions.

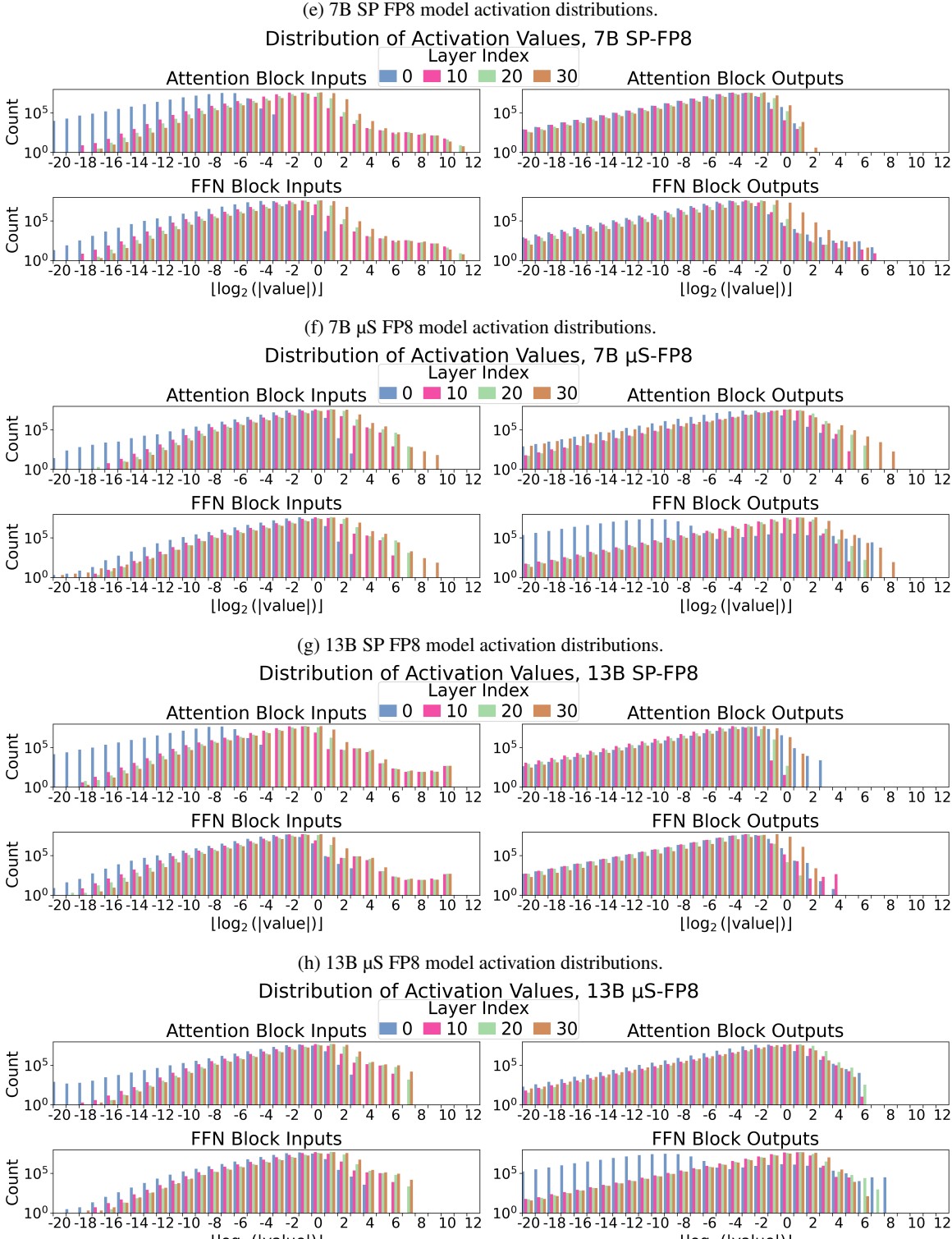

Figure 13. **Activation distributions of µS and SP models.** Activation distributions for attention and FFN block inputs and outputs are shown for 1B, 3B, 7B, and 13B FP8 models at every 10th layer. µS models lack the notable right tail of activation outliers in block inputs that SP models suffer from. This may make them easier to quantize.

