# OpenReview forum: "µnit Scaling: Simple and Scalable FP8 LLM Training"
_ICML.cc/2025/Conference — ICML 2025 poster_

### Official Review · Reviewer_GMxA · 2025-03-12

**Overall Recommendation:** 4

**Summary:**

The paper introduces a new 8-bit training method for LLMs, by keeping all tensors close to unit variance.  The method has fewer hyper-parameters than earlier methods, and is more straightforward to implement.  It also enables more accurate hyperparameter transfer from small models to large ones.

**Claims And Evidence:**

The main claims of the paper are well supported:
- Claim:  Proposed method achieves comparable accuracy to bf16 training.
Authors compare against equivalent models trained with bf16 at several scales (up to 13b) and show training loss is within ~1%.  Only thing to make this claim stronger would be to train for more steps.  Training steps are a bit short, but this is understandable and probably due to compute limitations.
- Claim: Proposed method is more practical than previous fp8 training methods
The method has only 3 hyperparameters which compare favorably to comparable methods, and these parameters also transfer better from small to large models, making the method more practical.  It also looks easier to get stable training with this method.  I'm conviced.

**Essential References Not Discussed:**

NA

**Experimental Designs Or Analyses:**

Experiment design is mostly straightforward as it should be.  The setup and presentation for the hyperparameter transfer results are rigorous and convincing.

**Methods And Evaluation Criteria:**

Metrics are training loss and standard LLM benchmarks, which is the right set of metrics here.  Training steps could be longer, if more compute is available, to make a stronger case.

**Other Comments Or Suggestions:**

NA

**Other Strengths And Weaknesses:**

NA

**Questions For Authors:**

What modifications induce the stability observed over SP?
Which method allows to reduce hyperparameters over \mu P?

**Relation To Broader Scientific Literature:**

This is the part I'm most unsure about, as I haven't followed the recent literature very closely.  Most of the methods presented (e.g. post-layer norm, unit variance initialization and modifications) have been used in prior work, and the authors cite those works where appropriate.  It is hard for me to judge, which changes introduced here result in the improvements over prior work like \mu P.  For instance table 1 is informative to show changes wrt standard transformer, a similar table would be very useful to compare against \mu P and SP as well.

For example what modifications induce the stability observed over SP?  Which method allows to reduce hyperparameters over \mu P

**Theoretical Claims:**

Theoretical claims look correct.

---

> ### Author Rebuttal · Authors · 2025-04-01
>
> We thank the reviewer for their detailed feedback on our work and are glad that they find the method’s theoretical basis and empirical results rigorous and compelling.
>
> ## Training duration
> > Only thing to make this claim stronger would be to train for more steps. Training steps are a bit short, but this is understandable and probably due to compute limitations.
>
> We agree with the reviewer that training for a longer duration would have been nice to make our claims even stronger. As the reviewer has anticipated, we had only a limited amount of compute and had to design our 1B, 3B, 7B, and 13B training runs accordingly. We are excited to see future work applying µS at even larger scales.
>
> ## Changes vs. SP and µP and their roles
> > For instance table 1 is informative to show changes wrt standard transformer, a similar table would be very useful to compare against \mu P and SP as well.
>
> We agree that denoting differences between µS and other training methods is useful. Table 1 in the paper shows changes that µS makes relative to the standard transformer (i.e., standard parametrization, or SP), which is the main comparison we aim to make. As a supplement to Table 1, we have prepared another table that enumerates differences between µS and other training methods (see table: https://imgur.com/a/f4dglps), which we plan to include in the appendix of the final paper.
>
> > What modifications induce the stability observed over SP?
>
> Numerical stability is a result of variance preservation in µS, making tensors better representable with low-precision formats. The subtle point we would like to emphasize is that variance preservation is an AND function; many components of µS all work in conjunction to achieve it. Unless the *entire* residual block preserves variance, the model doesn’t preserve variance. The entire model needs to be variance preserving in order to achieve better numerical stability over SP, and this is a conjunction of several modifications: linear layer scaling factors, post-branch-norm, fixed residual modification, and unit variance initialization.
>
> > Which method allows to reduce hyperparameters over \mu P?
>
> We need to tune fewer hyperparameters with µnit Scaling than µP because:
> - **Improved stability leads to more simplicity with µS**
>     - As discussed earlier, improved stability is the result of several components of µS together. Because training is more stable, we do not need to tune more hyperparameters to achieve reasonable performance with µS. Of course, if we added more hyperparameters and multipliers to tune it is reasonable to expect marginally better performance, but this is simply not necessary with the µS approach.
> - **Design choices eliminate extraneous hyperparameters**
>     - Enforcing near-unit variance in µS models by design eliminates hyperparameters related to initialization and individual layers. The examples below contrast hyperparameters tuned with µP (see Section F.4 of Tensor Programs V) with µS.
>         - µP tunes the weight initialization scale. µS initializes weights from $\mathcal{N}(0,1)$ by design.
>         - µP tunes the attention temperature. µS maintains queries and keys with near-unit variance by design and does not require this.
>         - µP tunes the embedding multiplier. µS keeps activations near-unit variance by design and avoids this.
>
> As a result, **training performant LLMs with µS only requires three hyperparameters**: learning rate, weight decay, and the residual coefficient.

---

### Official Review · Reviewer_NZBg · 2025-03-19

**Overall Recommendation:** 3

**Summary:**

This paper presents µnit Scaling(µS), a straightforward and scalable FP8 training method. It addresses the root causes of numerical instability in conventional transformer blocks and proposes effective solutions to mitigate these issues. µS approach incorporates Square-root Softmax Attention and Post-Branch-LayerNorm within transformer blocks, along with zero-shot hyperparameter transfer, enabling hyperparameters tuned on smaller networks to be applied directly to larger models. µS demonstrates stable training for LLMs up to 13B parameters in FP8 precision without dynamic scaling and achieves a 25-33% throughput increase compared to NVIDIA’s Transformer Engine.

**Claims And Evidence:**

Most of the key claims in the paper are well-supported by experimental evidence.

**Essential References Not Discussed:**

Most of the related works are cited and discussed in the paper.

**Experimental Designs Or Analyses:**

The experimental design effectively demonstrates µnit Scaling’s strengths in FP8 training stability, throughput, and hyperparameter transfer. However, limited benchmark diversity restricts insights into µS’s broader applicability.

**Methods And Evaluation Criteria:**

The proposed methods and evaluation criteria are well-suited to the core problem of FP8 training for LLMs. The design effectively addresses stability issues, improves efficiency, and reduces tuning costs. However, extending experiments to diverse architectures, tasks, and hardware platforms would provide stronger evidence of µnit Scaling’s robustness and versatility.

**Other Comments Or Suggestions:**

Summarizing the symbols and variable names used in the paper in the appendix can improve the readability of the article.

**Other Strengths And Weaknesses:**

Strengths:

1.	The paper is well-written with minimal typos.

2.	Figures int the paper are well designed and easy to understand.

3.	The introduction and demonstration of the method are sufficient.

Weaknesses:

1.	The layout of the article increases the difficulty of reading.

2.	Lack of experimental evidence demonstrating the necessity of different components of the method.

**Questions For Authors:**

1.	Are there ablation experiments on the role of different components of µS?

2.	Can µS be effective in models from other modalities or for different types of tasks?

**Relation To Broader Scientific Literature:**

Square-root Softmax Attention:

Conventional softmax attention mechanisms in Transformers are known to amplify large activations, which can destabilize FP8 precision due to overflows in matrix multiplication. The Square-root Softmax Attention in this paper offers a lightweight yet effective stability solution for FP8.

Post-Branch-LayerNorm:

Conventional Transformer architectures place LayerNorm before the residual connection. µS adapts post branch layernorm ideas specifically for FP8 precision, ensuring better variance control in deep LLMs.

Residual Modification Schemes:

Research on Deep Residual Networks has demonstrated that scaling residual branches can mitigate gradient explosion or vanishing gradients. This paper introduces fixed residual modification that stabilizes variance across deep Transformer layers.

**Theoretical Claims:**

The paper introduces a modification to the conventional softmax attention mechanism by applying a square root to the softmax scores. The provided proof establishing the relationship between sequence length and variance is sound, and the advantages of using Square-root softmax attention are effectively demonstrated through both theoretical analysis and experimental results.

---

> ### Author Rebuttal · Authors · 2025-04-01
>
> We appreciate the reviewer’s comments on our work, and are glad that the ideas we presented are clear.
>
> ## Ablating components of µS
> > Are there ablation experiments on the role of different components of µS?
>
> Most interventions in µS are **uniquely determined by simple math and the design goals** of:
> 1. Enforcing near-unit variance in all tensors
> 2. With negligible overhead
> 3. While enabling hparam transfer
>
> In the few cases where there are degrees of freedom, we either adhere to common best practices or perform ablations.
>
> We’ve added a new section at the start of the Appendix that spells out the origins of each component and we are modifying the text to make this clearer. An abbreviated version of this section is as follows:
> - **Unit variance init** - Necessary to ensure that weight tensors have unit variance.
> - **Linear layer static scales** - Since we don’t scale down the weights by $\frac{1}{\sqrt{\text{fan in}}}$, we have to scale linear layer outputs by this factor to maintain unit variance outputs.
> - **Learning rate scaling** - Based on µP, this is the unique way of scaling LR with model width that enables hparam transfer with the above weight initialization and static scaling factors.
> - **Weight decay scaling** - We adhere to the best practice of using decoupled weight decay from prior work, since coupled weight decay complicates hparam transfer. This is noted so our µS “recipe” is fully self-contained.
> - **FP8 hidden layers** - Standard practices for FP8 training, again included for completeness.
> - **Post-Branch-Norm** - As shown in Section 2.1, masked self-attention has diminishing variance with sequence position. This cannot be corrected with per-tensor scaling factors. Norm placement *can* correct this, so this degree of freedom has ablation results in Fig. 4b.
> - **“Fixed” residual modification** - Maintaining unit variance in the residual stream requires a weighted sum, but how to weight the branch and stream is a degree of freedom. We use the same weights across all residuals based on the results in Fig. 5.
>
> Omitting one or more of these modifications would prevent unit variance and/or hparam transfer, as shown by the mathematical analysis. While it may be interesting to explore scenarios where training doesn’t completely fail without some components, these results would not be useful enough to warrant inclusion–especially since these components are easy to implement and have minimal overhead.
>
> Another subtlety is that variance preservation and hparam transfer are AND functions. Unless the *entire* residual block preserves variance, the model will not. Similarly, weight init *and* static scales *and* learning rate must all work in conjunction to get hparam transfer. **The above modifications are not independent, additive tweaks–they are a minimal set to achieve the desired properties.**
>
> While the existing results already provide ample justification for the components of µS, to be completely sure that we address the reviewers’ comments, **we performed further experiments ablating norm placement and residual modification choices.**
> - Norm placement: Pre- vs. Post-branch norm with µS models (see https://imgur.com/a/qhq9CTP)
>     - Post-branch-norm converges better than pre-norm with µS in FP8. Supports the theoretical motivations for post-branch-norm from Section 2.1 (i.e., maintaining residual stream variance).
>     - Supplements our existing norm ablation in Fig. 4b.
> - Residual modification: Fixed, running-mean and standard residual modification (see https://imgur.com/a/eByqlYO)
>     - µS models that use standard residuals (i.e., no coefficients) do not properly converge. Supports the theoretical motivation for fixed residual modification from Section 2.2 (i.e., maintaining stream variance).
>     - Supplements our existing residuals ablation in Fig. 5.
>
> **In the final paper, we will compile all of these ablation results into a single subsection to clarify the contribution of the µS components. We hope this addresses the reviewers’ feedback about ablation studies.**
>
> ## Other modalities, architectures, and tasks
> > Extending experiments to diverse architectures, tasks, and hardware platforms would provide stronger evidence of µnit Scaling’s robustness and versatility.
>
> > Limited benchmark diversity restricts insights into µS’s broader applicability.
>
> > Can µS be effective in models from other modalities or for different types of tasks?
>
> We agree that applying our ideas to more modalities and architectures would be interesting. However, we believe improving LLM training is already enough scope to have a large, real-world impact. Further, nothing in µS is specific to H100s – µS is useful whenever properties listed in Fig. 1 are desirable.
>
> ## Article layout improvements
> > The layout of the article increases the difficulty of reading.
>
> If the reviewer could please elaborate on what we can improve, we are glad to address it.

---

### Official Review · Reviewer_F349 · 2025-03-19

**Overall Recommendation:** 2

**Summary:**

This paper introduces µnit Scaling (µS), a method for efficient FP8 training of large language models without requiring dynamic scaling factors or extensive hyperparameter tuning. µS builds on Unit Scaling to maintain unit variance in weights, activations, and gradients, ensuring stable low-precision training. It enables hyperparameter transfer across model sizes and eliminates the need for mixed-precision layers, allowing all hidden linear layers to compute in FP8. The method achieves training speedups of up to 33% while maintaining quality comparable to higher-precision baselines.

**Claims And Evidence:**

The claims made in this submission are generally supported by Fig 2, Fig 7, Table 5.

**Essential References Not Discussed:**

None

**Experimental Designs Or Analyses:**

The proposed µnit Scaling (µS) combines µP and Unit Scaling. However, the main results did not include these closely related baselines: µP, u-µP, and Unit Scaling.

**Methods And Evaluation Criteria:**

The evaluation criteria make sense for the problem and are aligned with previous papers.

**Other Comments Or Suggestions:**

None

**Other Strengths And Weaknesses:**

Strengths:

(1) It is interesting to see the theoretical analysis of the attention output variance.

(2) This paper is well-organized and easy to follow.

Weakness:

(1) The novelty is limited.  The proposed µnit Scaling (µS) scheme combines previously published µP and Unit Scaling.

(2) Lack of Baselines.  µP, u-µP, and Unit Scaling are strongly related techniques that should be included in the main results like Table 5 and Figure 7.

(3) Lack of Ablation studies. The proposed µnit Scaling (µS) scheme contains several modifications as shown in Table 1. However, there are no ablations to show each contribution of these modifications.

**Questions For Authors:**

(1)  Could you include µP, u-µP, and Unit Scaling baselines in your main results?

(2) Could you include ablation studies for the proposed µnit Scaling (µS) ?

**Relation To Broader Scientific Literature:**

https://arxiv.org/pdf/2407.17465

**Theoretical Claims:**

Did not completely check the correctness of the proofs.

---

> ### Author Rebuttal · Authors · 2025-04-01
>
> We greatly appreciate the reviewer’s helpful feedback and questions.
>
> ## Novelty of µS
> > The novelty is limited. The proposed µnit Scaling (µS) scheme combines previously published µP and Unit Scaling.
>
> While µS does build on ideas from both of these methods, µS involves modifications that are not present in either and obtains desirable properties that neither of these methods achieve (c.f. Figure 1: https://imgur.com/a/v1oA2cH). In particular, µS achieves hparam transfer with fewer extra hparams than µP and has better FP8 numerics at scale than Unit Scaling.
>
> We elaborate on specific differences below.
>
> ### µP
>
> µP enables hparam transfer from smaller to larger models. **However, µP suffers from numerical instabilities even with 16-bit formats**. As described in Section 7.4 of their paper, numerical issues caused frequent divergences that required them to train their GPT-3 in FP32. In contrast, µS provides hparam transfer even in FP8. µP also requires tuning many more hparams than µS (6 vs. 3, see Table 3), as well as changes such as $\frac{1}{d}$ attention and zero-initialization for some layers, which µS does not impose. **In summary, µS is a simpler solution than µP for hparam transfer on top of enabling FP8 training.**
>
> ### Unit scaling
>
> Unit Scaling facilitates low precision training by maintaining near-unit variance of weights, activations, and gradients through unit variance weight initialization and per-operation static scaling factors. However, as we show in Section 2.1, the masked self-attention operation central to LLMs has diminishing output variance over sequence position, **which Unit Scaling doesn’t address**. Our work is the first to identify this issue and uses post-branch-norm to address it. We demonstrate LLM training up to 13B parameters in FP8, while the largest model trained in the Unit Scaling work was 340 million params (BERT Large). µS also enables hparam transfer, which Unit Scaling does not. **In summary, µS fixes key numerical issues that Unit Scaling does not, enables hparam transfer, and scales FP8 training to much larger model sizes**.
>
> ### u-µP
>
> Concurrent work on u-µP also builds on both µP and Unit Scaling. However, **unlike µS, u-µP cannot keep all hidden layers in FP8**, requiring “critical matmuls” in transformer blocks to stay in BF16. u-µP also requires tuning many more hparams than µS (7 vs. 3, see Table 3). Key architectural modifications such as post-branch-norm and fixed residual coefficients permit µS to scale better. **In summary, unlike u-µP, µS provides full FP8 training for large LLMs and does so with fewer hparams.**
>
> ## Ablating components of µS
> > Lack of Ablation studies.
>
> Please refer to the “Ablating components of µS” section of our response to Reviewer NZBg.
>
> ## Additional baselines
> > Lack of Baselines. µP, u-µP, and Unit Scaling are strongly related techniques that should be included in the main results
>
> We have attempted comparison to Unit Scaling (US), but on models 7B and larger, US models were unable to converge in FP8 (see this representative figure at 7B model size: https://imgur.com/a/lJAMKmU). Further, our comparison of pre-norm vs. post-norm in FP8 showed that pre-norm (which US models use) had worse convergence (see here: https://imgur.com/a/qhq9CTP). In light of these findings, and because SP was more important to compare with, we allocated resources towards SP baselines at large scales instead of US.
>
> We did not compare to a µP baseline because of the difficulty of training µP models in 16 bit formats, let alone with FP8 (see “µP” subsection above). Being forced to train in FP32 means that 1) we can’t obtain an apples-to-apples comparison, and 2) collecting results for this baseline would be extremely expensive.
>
> We did not compare to a u-µP baseline since this work was done concurrently with our own. Given more time/compute resources, we agree that this comparison would be interesting, but unfortunately is not feasible to complete. Similar to our work, the u-µP work also only uses SP as a baseline.
>
> While assessing these additional baselines could be valuable, we note that **our ***demonstrated*** results are stronger than any existing method’s ***claimed*** results**. Even if we were to tune the hparams of these methods and achieve good hparam transfer and FP8 training quality, these methods would not surpass our µS results. This is because the focus of all methods, including ours, is on *qualitative* rather than quantitative properties–existing methods cannot fully preserve BF16 accuracy *even more* or keep the optimal hparams *more* unchanged with width. µS already matches BF16 quality with FP8 and achieves near-perfect hparam stability, leaving room for only tiny improvements on these axes. Such small improvements would not outweigh our method’s **reduced hyperparameter count**, **faster training**, and **alignment between training and inference precisions**.

---

### Decision · Program_Chairs · 2025-05-01

**Decision:**

Accept (poster)

**Comment:**

Reviewers generally agree that this is paper represents progress towards all-FP8 training of large models.
In comparison to existing work (mu-P, u-mu-P, unit scaling), the paper:
- maintains more weights in FP8
- uses post-branch-norm to address the sequence-dependence of output variance in masked attention
- provides hyperparameter transfer over more hyperparameters

Concerns about comparisons to existing work (some essentially concurrent) are well covered in the rebuttal.

A remaining concern is the small number of training steps - given the number of runs represented in this paper,
it would be extremely reasonable to ask for a realistic number of steps on at least one or two large scale runs.
Without seeing any indication of behaviour on longer runs, all of the above advantages cannot simply be assumed
to apply.